## Use of patient-held information about medication (PHIMed) to support medicines optimisation: protocol for a mixed-methods descriptive study

Sara Garfield,[1,2] Dominic Furniss,[3] Fran Husson,[1] Margaret Turley,[1] Bryony Dean Franklin[1,2]

¹Centre for Medication Safety and Service Quality, Imperial College Healthcare NHS Trust, London, UK
²UCL School of Pharmacy, London, UK
³UCLIC, University College London, London, UK

**Correspondence to**
Dr Sara Garfield;
sara.garfield@nhs.net

## ABSTRACT

**Introduction** Risks of poor information transfer across health settings are well documented, particularly for medication. There is also increasing awareness of the importance of greater patient activation. Patients may use various types of patient-held information about medication (PHIMed) to facilitate medication transfer, which may be paper or electronic. However, it is not known how PHIMed should best be used, whether it improves patient outcomes, nor is its key 'active ingredients' known. Discussion with patients and carers has highlighted this as a priority for research. We aim to identify how PHIMed is used in practice, barriers and facilitators to its use and key features of PHIMed that support medicines optimisation in practice.

**Methods and analysis** This study will take place in Greater London, England. We will include patients with long-term conditions, carers and healthcare professionals. The study has four work packages (WPs). WP1 involves qualitative interviews with healthcare professionals (n=16) and focus groups with patients and carers (n=20), including users and non-users of PHIMed, to study perceptions around its role, key features, barriers and facilitators, and any unintended consequences. WP2 will involve documentary analysis of how PHIMed is used, what is documented and read, and by whom, in a stratified sample of 60 PHIMed users. In WP3, we will carry out a descriptive analysis of PHIMed tools used/available, both electronic and paper, and categorise their design and key features based on those identified in WP1/2. Finally, in WP4, findings from WPs 1–3 will be integrated and analysed using distributed cognition as a theoretical framework to explore how information is recorded, transformed and propagated among different people and artefacts.

**Ethics and dissemination** The study has National Health Service ethics approval. It will provide initial recommendations around the present use of PHIMed to optimise patient care for patients, carers and healthcare professionals.

## INTRODUCTION
### The problem
The risks of poor information transfer across health settings are well documented,

### Strengths and limitations of this study

► A wide range of patient-held information about medication (PHIMed) tools will be considered including paper and electronic versions.
► Contextual factors relating to the use of PHIMed will be considered as well as the tools themselves.
► The study will have strong patient and public involvement including involvement in data analysis.
► A mixed-methods approach will allow triangulation of qualitative and quantitative data.
► A limitation is that the data collected will be descriptive, rather than statistically representative and limited to one urban geographical area.

particularly for medication.[1] It has been estimated that up to 60% of patients admitted to hospital have at least one discrepancy on their admission drug history.[2] While there are relatively few UK studies, in March 2007 the National Reporting and Learning System for England and Wales reported 7070 medication errors involving admission and discharge with 2 fatalities and 30 that caused severe harm.[3] Empirical studies suggest that in the hospital setting, prescribing errors are most likely at admission, largely due to challenges of medication reconciliation.[4 5] An audit of more than 8600 patients across 50 English hospital trusts found that when medicines were checked after admission, most patients had at least one omitted drug or wrong dose.[6] Earlier estimates suggested that between 30% and 70% of patients have either an error or an unintentional change to their medicines when admitted to hospital.[7] Problems are also common following transfer from hospital into the community[8 9] and when attending outpatient appointments.[10] A survey completed by 113 London general practitioners (GPs) to identify priorities for improvement of medication safety in primary care suggested

addressing incomplete reconciliation of medication as the highest priority.[11]

In the UK, medication prescribed by a patient's GP is generally listed on their electronic summary care record, which can increasingly be accessed by other healthcare professionals. Some aspects may also be viewable by patients. However, these records do not include over the counter or some specialist medication prescribed by hospitals, are sometimes inaccurate, and may have limited functionality.

Increasing patient (and carer) involvement with their medication records is a potential approach to improving information transfer across settings. There is increasing awareness of the importance of patient involvement and activation, where patient activation describes the knowledge, skills and confidence a person has in managing their own health and care, reflecting attitudes and approaches to self-management and engagement with healthcare.[12] It is widely recognised that people who feel in control, empowered and confident have better outcomes.[12] Supporting greater patient involvement is a fundamental component of 'person-centred care', a key feature of National Health Service (NHS) England's 5-year forward view[13] as well as being advocated by leading patient groups. A recent systematic review of carers' roles in preventing and facilitating medication errors in domiciliary settings shows that medication administration errors made by carers are a potentially serious patient safety issue and recommended better communication between carers and healthcare professionals.[14] It is therefore important to consider how to optimise patient and carer involvement in transferring medication-related information across care settings.

Many people who take medication use various types of patient-held information about medicines (abbreviated here to patient-held information about medication (PHIMed)). This may be paper or electronic, and may be based on formal documents from healthcare providers, coproduced between patients and healthcare professionals or informal documents created by patients themselves. There are a plethora of examples of such PHIMed available for use. Examples include My Medication Passport,[15] ThinkSafe,[16] Microsoft Health Vault,[17] the Lions Club International 'Message in a bottle',[18] 'My Medicine My Choice My Record' for care home residents,[19] Patients Know Best[20] and other electronic apps available within the Apple and Android app stores.

However, it is not known how PHIMed is used in practice, nor its key 'active ingredients' in terms of what it comprises and how it is used. As with many healthcare interventions, any benefits may arise due to the tool itself, or to the wider context, such as through the conversations and thought processes it stimulates and facilitates. This distinction has been referred to as the 'hard core' and the 'soft periphery' of an intervention.[21] Understanding the likely mechanism of action and an associated logic model[22] are therefore important to optimise PHIMed tools.

## The present need for research

Informal discussions with patients and carers within Northwest London highlighted transfer of information about medicines and the use of patient-held records as priority areas for research.

We, therefore, conducted a preliminary literature review in 2017 to identify and evaluate studies that have investigated the implementation, sustainability and/ or evaluation of PHIMed. We searched the databases EMBASE, PubMed, PsychLIT and the Cumulative Index to Nursing and Allied Health Literature, with search terms comprising the following: document*, medicine, medication passport, handheld, patient and medication passport. Studies that identified or evaluated a paper or electronic editable list of current medications to be carried by patients were included. Studies not published in English were excluded. Eleven studies were identified. Of these, three focused on the paper version of My Medication Passport,[15 23 24] one on another paper-based solution,[25] four on electronic solutions[26–29] and three on both paper and electronic solutions.[30–32] Collectively, these studies suggested that many patients brought some information about medicines with them to hospital, although this was rarely a formal document and there was little information about how it should be used or its key features. Suggested barriers to successful PHIMed implementation included confusion over who was responsible for updating it, lack of understanding as to its purpose, practicalities such as whether it fits into a pocket and lack of space to record potentially important details such as patient preferences for administration. A suggested facilitator was that any PHIMed met a clear need for potential users.

While this literature review identified a small number of barriers and facilitators, there has been no formal study of the barriers and facilitators to the use of PHIMed. For example, barriers may include patient and carer assumptions that since nearly 100% of GP surgeries and 69% of hospitals[33] use electronic prescribing, information on patients' medicines is automatically available to all healthcare professionals, the view that maintaining medication records is the role of healthcare professionals, previous reactions or discouragement by healthcare professionals, health literacy, concerns about inaccurate records, design barriers or information governance concerns. Facilitators may include encouragement by healthcare professionals, other patients or family members, a desire to take ownership and active involvement within healthcare, an organised approach to other aspects of health, and previous experience of the problems of fragmented medication records within health and social care.

Previous studies have generally taken a top–down approach of PHIMed being developed by healthcare professionals or app developers and then evaluated to investigate the extent to which people used it for its developed purpose. Only one study took a more bottom–up approach to developing PHIMed, which was for children with cystic fibrosis.[25] In our study, we will a take a

bottom–up approach in exploring the general needs of patients, carers and healthcare professionals in relation to PHIMed, the way in which they are currently using it and how they would like to do so.

This work will support the Royal Pharmaceutical Society of Great Britain's principles of medicines optimisation,[34] namely aiming to understand the patient's experience, evidence-based choice of medicines, ensuring medicines use is as safe as possible, and making medicines optimisation part of routine practice. The study is relevant to all of these, particularly the first, third and fourth principles. Our aims are to identify how PHIMed is used in practice, barriers and facilitators to its use, and key features of PHIMed that support medicines optimisation in practice, leading to the identification and development of effective PHIMed for testing in a future trial on health outcomes. Specific objectives are: (1) to explore perceptions of patients, carers and healthcare professionals around barriers and facilitators, benefits and unintended consequences of PHIMed; (2) to identify key PHIMed features likely to be required to support medicines optimisation; (3) to document how PHIMed is currently used in practice; (4) to describe PHIMed tools used/available within the UK, both paper and electronic, and the extent to which these provide the key features identified; (5) to inform development of a PHIMed solution for testing in a controlled trial on patient outcomes and (6) to make initial recommendations in relation to the current use and future development of PHIMed.

## METHODS AND ANALYSIS
### Overall study design
We will conduct a mixed-methods descriptive study. Qualitative interviews and focus groups will enable us to identify perceptions around the key components of successful and unsuccessful PHIMed; further qualitative and quantitative methods will then allow us to explore how PHIMed is used in practice and the extent to which current PHIMed solutions meet the requirements identified.

### Participants
We will include patients, carers and healthcare professionals from the Greater London area. Patients will be eligible to participate if they have had at least one long-term condition for at least 1 year, and take at least one prescribed medication. In order to recruit a diverse sample and to include those likely to have different needs in relation to PHIMed, we will conduct some focused recruitment in specific disease areas. For example, we will do focused recruitment of people with sickle cell anaemia as they tend to be younger and non-white, and of people with Parkinson's disease as they are likely to have specific requirements around brands, formulations and timing of medication.

### Definition of PHIMed
We define PHIMed as any patient-held information that allows for an editable list of current medications to be carried, regardless of whether or not other functionalities are also available. This could include both paper and electronic tools, including printed repeat medication lists, structured paper medication records and medication diaries. We exclude supplies of patients' own drugs (including those in compliance aids), drug-specific tools such as warfarin booklets, patient portals allowing read-only access to GP or hospital medical records, and medication reminder apps that do not support documentation of a list of current medication.

### Theoretical framework
Communication about medication involves information processing across people, tools and artefacts. Distributed cognition is a theoretical framework specifically designed to understand these kinds of sociotechnical systems.[35] It uses cognitive framing, based on information processing concepts, to explore interactions that are distributed across the members of a social group, across internal and external (material or environmental) structure, and over time.[36] For example, it has been used to explore propagation and transformation of information in cockpits,[37] how communication contributes to situation awareness in surgery,[38] as well as medication errors in care homes.[28] Distributed cognition for teamwork (DiCoT) is a framework that facilitates the application of distributed cognition in practice.[39 40] It helps consider different information flows within the system, how they are influenced by people and the design of tools and artefacts, how information is processed over physical spaces and how it evolves over time. DiCoT can be used to analyse interactions at the micro (individual), meso (team) and macro (organisational) levels.[41] Hence DiCoT seems ideal to explore the design and usability of PHIMed, how it helps or hinders fragmented healthcare communication, and how broader organisational contexts might affect its use and effectiveness.

DiCoT supports structured analysis in the form of five integrated models: an information processing model of information flow; a social model of the roles, skills and knowledge of the people involved; an artefact model that looks at design and usability of different tools and artefacts; a physical model that focuses on the spatial arrangement of equipment and information; and an evolutionary model that explores how the sociotechnical system evolves over time. For each model, schematic diagrams are used to represent details of information transfer and the implications for wider system performance. Each model also has a set of distributed cognition concepts and principles that serve as a vocabulary or checklist for analysing the model in terms of distributed cognition theory. DiCoT will therefore help explore how PHIMed: (1) supports patient cognition and interaction with healthcare professionals and (2) helps or hinders communication and coordination in the wider healthcare system.

The study has four work packages (WPs) all of which fit within the development phase of the Medical Research Council framework for evaluating complex interventions[42]; the piloting and evaluation phases will be addressed in a later study.

### WP1: exploring the context of PHIMed
#### Objectives
This WP will address study objectives 1 and 2.

#### Study design
Focus groups with patients and carers, and individual semistructured interviews with healthcare professionals.

#### Sampling
Twenty adult patients and adult carers of adults or children will be purposively sampled to include patients with and without carers, and both users and non-users of PHIMed, to represent a range of gender, ages, ethnicities and localities. Where adult patients have carers, they will be invited to bring their carer with them or to attend separately according to their preference. Some non-PHIMed users will be included to explore the reasons for not using PHIMed and any relevant barriers. We will hold two focus groups, each with 10 participants.

For healthcare professionals, we will use purposive sampling with the aim of creating a maximum variation sample (n=16) with respect to profession, gender, age, ethnicity, locality and previous experience with PHIMed. We envisage interviewing two GPs, two hospital doctors with experience of both inpatient and outpatient prescribing, two community pharmacists, two hospital pharmacists, two practice nurses, two hospital nurses, two dentists and two opticians.

#### Recruitment
Patient and carer participants will be recruited from primary and secondary healthcare organisations within the Greater London area, and relevant patient and carer groups and charities. We have already approached a range of collaboration partners to assist with recruitment. We will adapt our recruitment approach within each organisation to best fit the local context, using some or all of mailing, posters and direct approaches to potential participants in relevant clinical areas/waiting areas.

Healthcare professional participants will also be recruited from primary and secondary healthcare organisations within the Greater London area, using our personal and professional networks, as well as local clinical commissioning groups and the Royal Pharmaceutical Society local practice forums.

#### Data collection
Focus groups with patients/carers will be approximately 90 min in duration. The topic guide will include questions on the roles of PHIMed, why participants started using it, key features, barriers and facilitators to its use, and actual and potential unintended consequences (both positive and negative). Patients and parents/guardians will be invited to bring any PHIMed they use and we will (with their permission) make anonymised photos or sketches of materials provided. Focus groups will be digitally recorded and professionally transcribed; notes will also be taken to aid analysis and interpretation of the transcripts.

Individual healthcare professional interviews will be conducted either in person or via telephone depending on participant preference. Questions will include experiences around use of PHIMed such as what forms of PHIMed they have seen, who documents in PHIMed, how they respond when shown PHIMed, important features, perceived advantages and disadvantages of PHIMed, and both positive and negative unintended consequences. We will also explore the extent to which healthcare professionals and patients integrate PHIMed with diagnoses and medications listed on patients' hospital records and/or summary care records in primary care. Interviews will be approximately 30 min in duration and will be recorded and professionally transcribed.

#### Analysis
Transcripts will be analysed deductively with NVivo to support coding and analysis, using DiCoT as a framework. For example, DiCoT's social model will be used to outline how patients and carers perceive the healthcare system(s), the different professionals they interact with and communication between them. The information flow and artefact models will be used to explore the design of PHIMed and how it integrates with other healthcare systems. The evolutionary model will be used to explore what triggers use of PHIMed and how it evolves over time. Analysis will include searching for cases that contradict the main findings. A sample of 20% of transcripts will be analysed by two researchers as a reliability check. Lay partners will be involved in the analysis alongside professional researchers, an approach we have used previously.[43]

### WP2: exploring PHIMed use in practice
#### Objectives
This WP will address study objectives 2 and 3.

#### Study design
Individual semistructured interviews with PHIMed users.

#### Sampling
We will recruit a sample of 60 PHIMed users, including adults and adult carers of children, stratified according to use of digital versus paper PHIMed, and basic versus extensive PHIMed use to give 4 strata of 15 participants. 'Basic users' will be those keeping a list of current regular medications; 'extensive users' will be those also recording previous medications, short-term courses of medication and other information. Within each group, we will aim to include users with and without carers, as well as variation in age, gender, disease, ethnicity and number of medications. The sample size of 60 is based on 4 groups of 15 participants, where 10–15 is generally considered a suitable sample for qualitative research. Collectively, the

sample of 60 will also allow for a descriptive analysis of how PHIMed is used.

## Recruitment

Patient and parent participants will be recruited as for WP1.

## Data collection

Following recruitment and consent, we will ask participants to provide an overview of all health-related interactions over the last 3 months, using their diary or calendar as a reminder, together with details of interactions in which PHIMed was used, with whom, why and any reactions to its use. This will take place ideally in person, at a mutually convenient venue. We will ask those recruited to reflect on their experiences in the time between recruitment and data collection in order to aid recall. We will also ask participants how often they carry their PHIMed, what led them to start using it, whether and how they integrate it with their summary care record or other online NHS information via patient portals, and whether they have explored different PHIMed solutions. In addition, we will ask to view their PHIMed and ask questions to explore how up to date and comprehensive it is.

## Analysis

Notes and diagrams will be subject to thematic content analysis of how PHIMed is used, what is documented and read, and by whom. We will explore use of PHIMed as a cognitive artefact, in which the tool itself captures critical features of the issues it is trying to resolve and the deeper structures of individual and team cognition.[44] The DiCoT lens will allow us to explore the 'soft periphery'[21] of PHIMed, such as how users start thinking about medical conditions, medication and the integration of all this information, how they enter/record information and if necessary amend or delete it, and how they fashion the content in relation to their own experiences of usability and how it is perceived, received and used by healthcare professionals. Our stratified sampling strategy will also allow us to comment on the different affordances of paper versus electronic PHIMed. A 20% sample of the transcripts will be analysed by two professional researchers as a reliability check. Lay partners will be involved in the analysis alongside researchers as for WP1.

## WP3: features analysis of existing PHIMed solutions
### Objective
This WP will address study objective4.

### Study design
Descriptive quantitative analysis.

### Data collection
We will collate a list of desired features based on WP1/2, relevant literature and discussion with key stakeholders. Such features might include the ability to record current and recent medication (including over the counter and complementary medication, oxygen and vaccinations where relevant), allergies and sensitivities, compliance aids, problems with taking medication (eg, swallowing difficulties), ability to record comments about each medication, accessibility of information in an emergency and for electronic solutions, any certification or approval, such as with the NHS Health Apps Library.[45] A list of PHIMed tools will be obtained from a systematic search on the Apple app store, Google Play, Google, Pinterest, websites of patient charities such as Age UK and Patients Association, plus those already known to the research team, identified in WP1/2 or through literature review.

### Analysis
We will conduct a descriptive quantitative overview of PHIMed solutions used in the UK and will map these against the list of key features, with input from our lay partners into whether usability-related requirements are met. This will allow us to establish the proportion of tools that have each of these features. Inter-rater reliability will be assessed using the kappa statistic. The resulting findings will largely focus on enriching DiCoT's artefact model, while also considering the significance of features within the wider sociotechnical system.

## WP4: integrated analysis
### Objective
This WP will address study objectives 5 and 6.

Findings from WPs 1–3 will be integrated and analysed using a deductive approach, with DiCoT as a theoretical framework to explore how information is recorded, used and transformed among different people and artefacts. DiCoT's five models will be used to integrate these data to identify important patterns of structure and behaviour in this sociotechnical system, allowing opportunity for incremental and more radical design considerations and recommendations[46]: First, the information flow model will explore interactions between patients, carers and clinicians and how PHIMed acts as an information hub. Second, the artefact model will focus on PHIMed as a cognitive artefact, as well as considering other tools and artefacts used in conjunction. Third, the social model will look at the nature of the healthcare networks the patient interacts with, and their perceived fragmentation or cohesiveness. We anticipate that the social model will be relevant for understanding how PHIMed supports knowledge sharing and decision-making of people with different roles in this system, such as healthcare professionals, carers and patients. Fourth, the evolutionary model will look at what triggers patients' use of PHIMed, how it evolves over time and the role of coproduction between patients and healthcare professionals. Finally, the physical model will attend to how PHIMed is used within and between different physical spaces. For example, we will explore the portability of PHIMed, how it is stored and used in the home and when out and about, and how it can change the physical dynamics of patient consultations.

## Patient and public involvement

The research area arose through discussions with a number of patient and public groups. Two patients are co-authors on this protocol and are part of the research team. Four additional lay partners are members of the advisory group. They have been involved in developing patient facing materials and will be involved in management of the research, recruitment, data analysis and dissemination of research findings.

## ETHICS AND DISSEMINATION

We will invite informed consent from all participants prior to interviews and focus groups. All quotations and copies of PHIMed will be anonymised.

We plan to present our work at a suitable UK conference and publish at least one peer-reviewed research paper. In addition to this, we will produce summaries of our work. One will be in plain English and aimed at the public giving guidance on how best to use PHIMed. Another will be aimed at healthcare professionals to inform the use of PHIMed.

We will also produce initial guidance aimed at policy-makers to inform future development of PHIMed. All of the summaries will be made available via the websites of our affiliated organisations, and further disseminated via Twitter.

**Contributors**  SG and BDF led on study design and protocol writing with assistance from the other authors. DF wrote the sections relating to the use of DiCoT. FH and MT were actively involved in discussions leading to this proposal and have had substantial input into the scope of the proposal.

**Funding**  This work was supported by Pharmacy Research UK grant number PRUK-2016-PG2-2-A and by the National Institute for Health Research (NIHR) Imperial Patient Safety Translational Research Centre. This report presents independent research funded by Pharmacy Research UK (PRUK).

**Disclaimer**  The views expressed in this publication are those of the authors not necessarily those of PRUK, the NHS, the NIHR or the Department of Health.

**Competing interests**  None declared.

**Patient consent**  Not required.

**Ethics approval**  This proposal has been approved by East Midlands—Nottingham 2 Research Ethics Committee ref (17/EM/0477).

**Provenance and peer review**  Not commissioned; externally peer reviewed.

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
