## [Reviewer comments · BMJ Open]

ARTICLE DETAILS

TITLE (PROVISIONAL)	The use of Patient-Held Information about Medication (PHIMed) to support medicines optimisation: protocol for a mixed methods descriptive study
AUTHORS	Garfield, Sara; Furniss, Dominic; Husson, Fran; Turley, Margaret; Dean Franklin, Bryony

VERSION 1 – REVIEW

REVIEWER	Olaf Krause, M.D. Centre for Medicine of the Elderly DIAKOVERE Henriettenstift Hannover and Institute for General Practice Hannover Medical School (MHH), Germany
REVIEW RETURNED	22-Feb-2018

GENERAL COMMENTS	Dear authors, Thank you very much for your very interesting draft. The draft contributes to a very important topic, i.e. patient-held information, a problem every doctor knows by heart. The language of the draft is of course of high standard as it's written by native speakers. Nevertheless, it's easy to understand with a sound introduction. But there are some topics to be regarded and corrected: 1) The key words differ on the very first dashboard and on page 1 of the draft. This has to be corrected.2) Participants: The choice of participants is not logical to the reviewer and very probable to the future reader. Why patients with Parkinson's disease and sickle cell anaemia? What kind of cancer patients? Why do the authors not focus on common diseases like cardiovascular disease or diabetes mellitus? This has to be explained.3) The title of the draft isto support medicines optimisation: protocol.. How is this optimisation measured? It might be about reducing polypharmacy oder PIM or others objectives. This has to be added and precisely described.4) The study will take place in a very urban area. The use of PHIMed might differ totally e.g. in rural areas like Scotland or the Midlands. This is a very clear limitation and has to be pointed out obviously. The draft has its strengths (good readable, important topic, study design), but some weaknesses have to be considered. The manuscript could be accepted, given that the made reviewer's remarks are implemented.
---

REVIEWER	Afonso M Cavaco Faculty of Pharmacy, University of Lisbon, Portugal
-----------------	--

GENERAL COMMENTS

Thank you for the opportunity to read this very well written and detailed study protocol. My comments are as follows.

I would start by recommending to expand the study limitation in the Strengths and Limitations section. According to the proposed protocol, it is not assured the findings can be extrapolated also within the geographical area studied i.e. there are no statistical methods for results representativeness amongst the wider population from the same region.

Authors start their protocol description by mentioning information transfer across health settings. This is a concept widely accepted, but a precise initial definition would be beneficial for an international readership: are authors referring to an electronic patient record shared by communicating IT system in different providers? A NHS portal with patient access? Is it a paper-based information transfer through patients and carers? This is elucidated later, say in page 6 lines 19-21 and lines 47-54, but I believe an earlier explanation of the concrete circuit of information transfer would benefit all readers. Authors mentioned in page 5 "a plethora of" PHIMed resources, from official NHS to informal documents created by patients themselves. At the end of page 5, authors disclose their aim: to understand the likely mechanism of action and associated logical model of the existing PHIMed. This immediately rises the concern of how to analyse such quantity and diversity of PHIMed materials. For instance, at this point, I was not so sure if and how personal structured notes and medication diaries were to be included.

The specific study objectives appear to have a numerical order. Only as a suggestion, I would invite the authors to rethink the sequence. For instance, objectives 1 and 2 will be addressed by a qualitative approach, which does not guarantee to cover all PHIMed that are being used by patients at the study location. The identification of key PHIMed solution and features (objective 3) first would allow to have a discussion base for the objectives 1 and 2. Setting the scope for PHIMed that are documented as more prevalent in the location, would add to study feasibility in my opinion.

Two frameworks are mentioned, the DiCoT and the MRC for complex interventions. The 1st provides the main constructs, the interfaces and interactions between them, thus helping to analyse and make sense of the data. The 2nd is briefly mentioned (development phase) to support the study organizational approach in the proposed WPs. If this is right and thinking of the wider readership, I would recommend having a clearer role for both frameworks.

I would strongly recommend having a link between each WPs and the specific study objective to which the WP responds e.g. to restate objectives at the beginning of each WPs e.g. before study design.

In page 8 lines 25 to 27, I was wondering at the specific reasons to focus on these groups, besides convenience. The selection criteria seem inconsistent, being centred both in socio-demographics and in specific diseases/conditions, the latter already covered by some demographics.

WP1 presents a closed number of participants previewed for qualitative research methods, particularly for individual interviews with professionals. This might be based on authors previous experience with data redundancy, but it seems to me that an open sample size & recruitment would benefit the descriptive exploration of the subject matter.

I was confused by the WP2 title. To me this would be equivalent to a

	document analysis, in which documents are interpreted by the researcher to give voice and meaning around the assessment topic (Bowen, 2009). The WP2 main objective is to have a descriptive analysis of how PHIMed is used in practice. This seems already somewhat covered by the previous WP1. Here authors plan using a total of 60 participants, increasing for sure the robustness of the qualitative findings. However, a well-designed survey (including open questions and e.g. photo/sketches files upload) might respond better to the objectives and establishing the true mixed-methods design mentioned. In WP3 it was not clear to me how the 2 procedures contribute to analyse the features of existing PHIMed. With the previous qualitative WPs authors can't assure that all or the more prevalent PHIMed resources were found and characterised. Of course, most will be covered, if not all PHIMed being used, but this will be hard to demonstrate with confidence. Moreover, the systematic search of apps and websites should be validated with data from the actual usage/coverage of such resources in statistically representative population groups. Authors should also present how a collation of a list of features, from WP1/2 plus literature review plus discussion with key stakeholders, contributes to the analysis of the existing PHIMed. Will authors establish a minimum number of essential features, using e.g. a rating instrument? Will authors value innovating or disruptive options and how? In this sense, some sort of consensus should be included here, maybe from WP1 and/or 2. Authors mentioned in WP3 analysis an inter-rater reliability testing, while working with lay partners input. I find this difficult to achieve without defining and using e.g. a rating tool. Regarding WP4 it would be important to present how PHIMed relates to the social model of the DiCoT framework, as well as within the DiCoT physical model. I would reinforce the details concerning the ethical aspects of each WP, placing indications of participants voluntary consent and other ethical requirements, as part of the compliance with best healthcare research principles and practices. Thank you.
--	---

VERSION 1 – AUTHOR RESPONSE

bmjopen-2018-021764

The use of Patient-Held Information about Medication (PHIMed) to support medicines optimisation: protocol for a mixed methods descriptive study

Reviewers' Comments to the Author	RESPONSES
Reviewer: 1 Reviewer Name: Olaf Krause, M.D. Institution and Country: Centre for Medicine of the Elderly DIAKOVERE Henriettenstift Hannover and Institute for General Practice Hannover Medical School (MHH), Germany	
The draft contributes to a very important topic, i.e.	We thank the reviewer for these comments.

patient-held information, a problem every doctor knows by heart. The language of the draft is of course of high standard as it's written by native speakers. Nevertheless, it's easy to understand with a sound introduction. But there are some topics to be regarded and corrected: 1) The key words differ on the very first dashboard and on page 1 of the draft. This has to be corrected.	We have addressed these issues point by point below. We have now ensured that the key words are the same in both lists.
2) Participants: The choice of participants is not logical to the reviewer and very probable to the future reader. Why patients with Parkinson's disease and sickle cell anaemia? What kind of cancer patients? Why do the authors not focus on common diseases like cardiovascular disease or diabetes mellitus? This has to be explained.	We thank the reviewer for highlighting this need for clarification; we have now made this aspect clearer in the paper. Any patient who has had at least one long term condition for at least one year, and takes at least one prescribed medication will be eligible for inclusion. However in order to recruit as diverse a sample as possible and to include those likely to have different needs in relation to PHIMed, we are doing some focused recruitment in the areas listed. For example, people with sickle cell anaemia tend to be younger non-white patients, those with Parkinson's disease are likely to have specific requirements around brands, formulations and timing of medication, etc. Many of these groups will also be taking medication for common diseases such as cardiovascular disease or diabetes mellitus as the reviewer suggests.
3) The title of the draft isto support medicines optimisation: protocol.. How is this optimisation measured? It might be about reducing polypharmacy oder PIM or others objectives. This has to be added and precisely described.	This work will support the British Royal Pharmaceutical Society's principles of medicines optimisation:  1. Aiming to understand the patient's experience. 2. Evidence based choice of medicines 3. Ensuring medicines use is as safe as possible 4. Make medicines optimisation part of routine practice The study is relevant to all four principles,; particularly one, three and four. We have now added this to the paper.

	We will not be measuring optimisation at this developmental stage. We would envisage measuring patient outcomes such as adherence in a future evaluation but this is out of scope for the present paper.
4) The study will take place in a very urban area. The use of PHIMed might differ totally e.g. in rural areas like Scotland or the Midlands. This is a very clear limitation and has to be pointed out obviously.	We had acknowledged that the study will be limited to one geographical area but have now further clarified that this is an urban area.
The draft has its strengths (good readable, important topic, study design), but some weaknesses have to be considered. The manuscript could be accepted, given that the made reviewer's remarks are implemented.	We have addressed the reviewer's specific points as outlined above.
Reviewer: 2 Reviewer Name: Afonso M Cavaco Institution and Country: Faculty of Pharmacy, University of Lisbon, Portugal	
Thank you for the opportunity to read this very well written and detailed study protocol. My comments are as follows. I would start by recommending to expand the study limitation in the Strengths and Limitations section. According to the proposed protocol, it is not assured the findings can be extrapolated also within the geographical area studied i.e. there are no statistical methods for results representativeness amongst the wider population from the same region.	We thank the reviewer for these positive comments and are pleased he found our protocol to be well written and detailed. We had already stated that the study will be descriptive, but we have also now further specified that it will not be statistically representative of the wider population, in order to address this comment.
Authors start their protocol description by mentioning information transfer across health settings. This is a concept widely accepted, but a precise initial definition would be beneficial for an international readership: are authors referring to an electronic patient record shared by communicating IT system in different providers? A NHS portal with patient access? Is it a paper-based information transfer through patients and carers? This is elucidated later, say in page 6 lines 19-21 and lines 47-54, but I believe an earlier explanation of the concrete circuit of information transfer would benefit all readers.	We have now added more information about medication records in the UK and have placed this information earlier in the paper as suggested.
Authors mentioned in page 5 "a plethora of" PHIMed resources, from official NHS to informal documents created by patients themselves. At the end of page 5, authors disclose their aim: to understand the likely mechanism of action and associated logical model of the existing PHIMed. This immediately rises the concern of	We had defined PHIMed in the methods section. We have now clarified that this could include personally structured medication records and medication diaries. Our data collection methods are intended to capture and describe the anticipated diversity in types of PHIMed, as

how to analyse such quantity and diversity of PHIMed materials. For instance, at this point, I was not so sure if and how personal structured notes and medication diaries were to be included.	highlighted by the reviewer.
The specific study objectives appear to have a numerical order. Only as a suggestion, I would invite the authors to rethink the sequence. For instance, objectives 1 and 2 will be addressed by a qualitative approach, which does not guarantee to cover all PHIMed that are being used by patients at the study location. The identification of key PHIMed solution and features (objective 3) first would allow to have a discussion base for the objectives 1 and 2. Setting the scope for PHIMed that are documented as more prevalent in the location, would add to study feasibility in my opinion.	We have now changed the order of objectives 2 and 3 as suggested.
Two frameworks are mentioned, the DiCoT and the MRC for complex interventions. The 1st provides the main constructs, the interfaces and interactions between them, thus helping to analyse and make sense of the data. The 2nd is briefly mentioned (development phase) to support the study organizational approach in the proposed WPs. If this is right and thinking of the wider readership, I would recommend having a clearer role for both frameworks.	We have now added further information about our use of the MRC framework to make it clear to a wider readership that this is an established overarching evaluation framework.
I would strongly recommend having a link between each WPs and the specific study objective to which the WP responds e.g. to restate objectives at the beginning of each WPs e.g. before study design.	We have now referred to the relevant objectives at the beginning of each work package as suggested.
In page 8 lines 25 to 27, I was wondering at the specific reasons to focus on these groups, besides convenience. The selection criteria seem inconsistent, being centred both in socio-demographics and in specific diseases/conditions, the latter already covered by some demographics.	Please see our response to the similar point made by reviewer 1 above.
WP1 presents a closed number of participants previewed for qualitative research methods, particularly for individual interviews with professionals. This might be based on authors previous experience with data redundancy, but it seems to me that an open sample size & recruitment would benefit the descriptive exploration of the subject matter.	We agree that this would have been ideal, but for planning and funding purposes we needed to define our sample size in advance. We have used a maximum variation sampling strategy to include a broad range of viewpoints.
I was confused by the WP2 title. To me this would be equivalent to a document analysis, in which documents are interpreted by the researcher to give voice and meaning around the assessment topic (Bowen, 2009). The WP2 main objective is to have a descriptive analysis of how PHIMed is used in practice. This seems already somewhat covered by the previous WP1. Here authors plan using a total of 60 participants, increasing for sure the robustness of the qualitative findings. However, a well-designed survey (including open	In WP1 we will explore perceptions of patients, carers and healthcare professionals around barriers and facilitators, benefits and unintended consequences of PHIMed. In WP2 we will document how PHIMed is actually used in practice. We have now made this clearer by rewording the WP1 and WP2 titles, and by mapping the study objectives onto the WPs as suggested by the reviewer above.

questions and e.g. photo/sketches files upload) might respond better to the objectives and establishing the true mixed-methods design mentioned.	Documentary analysis is a qualitative research method and is therefore better suited to being used in the context of face to face interviews rather than using a survey methodology. Surveys also have the disadvantage of tending to have low response rates and less diversity among the respondents.
In WP3 it was not clear to me how the 2 procedures contribute to analyse the features of existing PHIMed. With the previous qualitative WPs authors can't assure that all or the more prevalent PHIMed resources were found and characterised. Of course, most will be covered, if not all PHIMed being used, but this will be hard to demonstrate with confidence. Moreover, the systematic search of apps and websites should be validated with data from the actual usage/coverage of such resources in statistically representative population groups.	The systematic search would be expected to find formal types of PHIMed that have been used. To complement this, WP1 and WP2 would be expected to provide examples of less formal types of PHIMed tools that may have been created by patients themselves. We have now made this clearer in the paper. Assessing the usage/coverage of PHIMed in statistically representative populations does not form part of our study's aim and objectives and is beyond the scope of the present study.
Authors should also present how a collation of a list of features, from WP1/2 plus literature review plus discussion with key stakeholders, contributes to the analysis of the existing PHIMed. Will authors establish a minimum number of essential features, using e.g. a rating instrument? Will authors value innovating or disruptive options and how? In this sense, some sort of consensus should be included here, maybe from WP1 and/or 2.	We have now added further detail about the WP3 analysis. Our aim in WP3 is to establish the proportion of tools that include the key features identified by different groups of PHIMed users, rather than to rate individual tools. We have now clarified this in the paper. We will search for cases that contradict the main findings in our analysis, as would be standard practice in qualitative research. We have now specified this in the analysis section.
Authors mentioned in WP3 analysis an inter-rater reliability testing, while working with lay partners input. I find this difficult to achieve without defining and using e.g. a rating tool.	The inter-rater reliability will relate to whether or not each of the key features in terms of user requirements are met for each PHIMed tool. We have now clarified this in the paper.
Regarding WP4 it would be important to present how PHIMed relates to the social model of the DiCoT framework, as well as within the DiCoT physical model.	In WP4 we had described how all five DiCoT models will be related to PHIMed, namely the information flow model, artefact model, social model, evolutionary model and physical model. We anticipate that the social model will be relevant in terms of understanding the healthcare networks that the patient interacts with and how PHIMed

	fits into these interactions, and that the physical model will be relevant in terms of how PHIMed is used within and between different physical spaces. We have now added slightly more detail to expand on these points in the paper.
I would reinforce the details concerning the ethical aspects of each WP, placing indications of participants voluntary consent and other ethical requirements, as part of the compliance with best healthcare research principles and practices	We have now added details of informed consent and confidentiality to the ethics section.